# Berberine-Loaded Liposomes for the Treatment of *Leishmania infantum*-Infected BALB/c Mice

**DOI:** 10.3390/pharmaceutics12090858

**Published:** 2020-09-09

**Authors:** Alba Calvo, Esther Moreno, Esther Larrea, Carmen Sanmartín, Juan Manuel Irache, Socorro Espuelas

**Affiliations:** 1ISTUN Institute of Tropical Health, University of Navarra, Irunlarrea 1, 31008 Pamplona, Spain; acalvo.5@alumni.unav.es (A.C.); emorenoa@unav.es (E.M.); elarrea@unav.es (E.L.); sanmartin@unav.es (C.S.); 2Chemistry and Pharmaceutical Technology Department, University of Navarra, Irunlarrea 1, 31008 Pamplona, Spain; jmirache@unav.es; 3IdisNA, Navarra Institute for Health Research, 31008 Pamplona, Spain

**Keywords:** berberine, liposomes, visceral leishmaniasis, *Leishmania infantum*

## Abstract

Berberine (BER)—an anti-inflammatory quaternary isoquinoline alkaloid extracted from plants—has been reported to have a variety of biologic properties, including antileishmanial activity. This work addresses the preparation of BER-loaded liposomes with the aim to prevent its rapid liver metabolism and improve the drug selective delivery to the infected organs in visceral leishmaniasis (VL). BER liposomes (LP-BER) displayed a mean size of 120 nm, negative Z-potential of −38 mV and loaded 6 nmol/μmol lipid. In vitro, the loading of BER in liposomes enhanced its selectivity index more than 7-fold by decreasing its cytotoxicity to macrophages. In mice, LP-BER enhanced drug accumulation in the liver and the spleen. Consequently, the liposomal delivery of the drug reduced parasite burden in the liver and spleen by three and one logarithms (99.2 and 93.5%), whereas the free drug only decreased the infection in the liver by 1-log. The organ drug concentrations—far from IC_50_ values— indicate that BER immunomodulatory activity or drug metabolites also contribute to the efficacy. Although LP-BER decreased 10-fold—an extremely rapid clearance of the free drug in mice—the value remains very high. Moreover, LP-BER reduced plasma triglycerides levels.

## 1. Introduction

Leishmaniasis encompasses a variety of pathologies caused by different species of the protozoan parasite *Leishmania* and spread by the bite of sand flies. The World Health Organization (WHO) classifies this disease to be prominent among the causes of death by infectious diseases worldwide, with 12 million infected people in 98 endemic countries [1]. Clinical manifestations produced by *Leishmania* include tegumentary and visceral forms. Regarding tegumentary forms, cutaneous leishmaniasis (CL) is the most common type of leishmaniasis, but CL infections are mild or asymptomatic in many cases. However, visceral leishmaniasis (VL) is the most severe form of the disease, affecting internal organs, in particular the spleen, liver and bone marrow (BM). It is highly fatal if untreated and responsible for 50,000 to 90,000 cases in 2017 [2].

To date, there are no effective vaccines and chemotherapy is the sole choice to treat this disease. For decades, antimonial derivatives have been the reference drugs for treating VL. However, these toxic compounds exhibit a narrow therapeutic window and resistance has become widespread. Alternative drugs such as miltefosine, paromomycin or amphotericin B are also associated with toxicity, high costs, long-term regimens and emergence of resistance [3]. Thus, the development of novel treatments against VL is an urgent need.

Berberine chloride (BER, 1,8,13-α-tetra-hydro-9,10-demethoxy-2,3-(methyl-ene-dioxy)-berberium chloride), an anti-inflammatory quaternary isoquinoline alkaloid present in medicinal plants (*Berberis vulgaris* and *Berberis aristata*), has been reported to have a variety of biologic effects [4,5], including antileishmanial activity. BER and different extracts containing BER exhibited activity in healing CL lesions [6,7]. Moreover, BER was effective in treating VL in vitro [8,9]. It has been shown to mediate its antileishmanial activity in *L. donovani* promastigotes and amastigotes by inducing a redox imbalance following enhanced generation of reactive oxygen species (ROS) and concomitant depletion of nonprotein thiols, which culminated in a caspase-independent apoptotic-like death [9,10]. Moreover, mitochondrial superoxide caused depolarization, inhibited electron transport chain and depleted ATP, thus, leading to apoptosis. Furthermore, BER also activated macrophages inducing iNOS and IL12p40 upregulation and finally NO production (essential to kill intracellular amastigotes), in both uninfected and *L. donovani*-infected macrophages [10]. In vivo, the antileishmanial activity of BER was tested in *L. donovani*-infected golden hamsters with a parasite burden reduction of 48.5 and 61.1% after 50 or 100 mg/kg intraperitoneally (i.p.) administered for four days, or a reduction of one logarithm (by 90%) in liver and spleen, when treated for 10 days at 50 mg/kg/day [11]. In another study, BER administered intramuscularly (i.m.) against *L. donovani*-infected hamsters at a total dose of 52 mg/kg caused a 20% of suppression, whereas a dose of 208 mg/kg had an infectiveness reduction of 36% in livers [12]. This BER mild, but improvable, in vivo leishmanicidal activity could be explained by its rapid metabolism and excretion [5] and its inadequate tissue distribution to target *Leishmania* parasites. Thus, to circumvent these problems, in the last years, the development of different delivery systems (DDS) containing BER such as albumin-based nanoparticles [13], polymeric nanoparticles [14], self-micro/nanoemulsifying drug delivery systems [15,16], micelles [17], liposomes [18] or solid lipid nanoparticles [19] for different applications such as cancer [18], cardiac dysfunction [18] or metabolic diseases [19], resulted in increased efficacy compared to the free drug [20]. However, there are no studies, up to our knowledge, in the development of DDS containing BER for the treatment of leishmaniasis. Regarding VL disease, many different DDS have been reported to target the drug to *Leishmani*a-infected organs, reducing the parasitic load and enhancing the therapeutic outcome [21]. Liposomes are considered appropriate carriers to treat VL for their natural tendency to be taken up by the mononuclear phagocyte system, namely by macrophages which are the cells where *Leishmania* parasite resides [22]. For many years, Amphotericin B has been used in clinical therapy for leishmaniasis in the form of a safe liposomal formulation known as AmBisome^®^ [23]. Moreover, promising approaches have been done in VL with other antileishmanial drugs like meglumine antimoniate [22,24] or paromomycin [25] encapsulated in liposomes.

Thus, this study aimed to evaluate the potential use of BER-containing liposomes for the treatment of VL. For this purpose, LP-BER were developed and characterized. Their pharmacokinetic profile was determined, and their antileishmanial activity was assessed in a *L. infantum*-infected mice model. This in vivo model is considered appropriate to first assay the efficacy of new drugs or treatments. If the results show a promising potential, they could be further evaluated in a hamster model, which best resemble the infection in humans [26].

## 2. Materials and Methods

### 2.1. Material

Berberine chloride (BER), amphotericin B (AmB), dimethyl sulfoxide (DMSO), 3-[4,5-dimethylthiazol-2-yl]-2,5-diphenyltetrazolium bromide (MTT), 4-(2-hydroxyethyl)piperazine-1-ethanesulfonic acid sodium salt (HEPES), adenine, hemin, biotin, ethylenediaminetetraacetic acid (EDTA), cholesterol (Chol), dicetyl phosphate (DP), didodecyldimethylammonium bromide (DDAB), d-α-tocopherol succinate (TS) and d-(+)-trehalose dihydrate were obtained from Sigma (St. Louis, MO, USA). Soybean phosphatidylcholine (PC) was kindly gifted by Lipoid GmbH (Ludwigshafen, Germany). Fungizone^®^ (FGZ) was obtained from X-GEN Pharmaceuticals, Inc. (Horseheads, NY, USA). All other reagents were of analytical grade.

### 2.2. Parasites

*Leishmania infantum* promastigotes (BCN-150) were maintained in flasks at 26 °C in continuous stirred M199 1X medium (Sigma, St Louis, MO, USA) supplemented with 10% heat-inactivated fetal bovine serum (FBS) (Gibco, Gaithersburg, MD, USA), 25-mM HEPES (pH 7.2), 0.1-mM adenine, 0.0005% (*w*/*v*) hemin, 0.0001% (*w*/*v*) biotin and 100 UI/mL penicillin and 100-mg/mL streptomycin (Gibco, Gaithersburg, MD, USA). Media was changed each day to achieve exponential growth. Parasites were also maintained in Schneider’s modified medium (Sigma, St Louis, MO, USA) supplemented with 20% FBS and 100 UI/mL penicillin and 100-mg/mL streptomycin for 5–6 days in order to obtain stationary cultures for the infection of macrophages and animals.

### 2.3. Obtainment and Culture of Bone Marrow-Derived Macrophages (BMDM)

Bone marrow-derived macrophages (BMDM) were isolated from BALB/c mice by flushing the femur and tibia with phosphate-buffered saline (PBS, Gibco, Gaithersburg, MD, USA). Bone marrow cells were then resuspended in Dulbecco’s modified Eagle’s medium (DMEM, Gibco, Gaithersburg, MD, USA) supplemented with 10% FBS, 1% penicillin/streptomycin and 20% filtered supernatant from L929 cell line (obtained from the ATCC collection, C3H/An genetic background), as source of granulocyte-macrophage colony stimulating factor (GM-CSF), required to induce hematopoietic cell differentiation into macrophages. Cells were incubated for 8 days at 37 °C and 5% CO_2_ with medium change every 3 days.

### 2.4. Animals

BALB/c mice (Harlan, Spain) weighing approximately 20 g were kept under conventional conditions with free access to food and water. Animals were housed in groups of 5 in plastic cages in controlled environmental conditions (12:12-h light/dark cycle and 22 ± 2 °C). This study was conducted according to ethical standards approved by the Animal Ethics Committee of the University of Navarra in strict accordance with the European legislation in animal experiments (protocol code number 100-19 approved by Government of Navarra on 18 December 2019). Animals were infected by intravenous (i.v.) inoculation of 10^8^ stationary-phase promastigotes of *L. infantum* in the tail vein.

### 2.5. Preparation of BER-Containing Liposomes

BER liposomes were obtained by the thin-film hydration method [27]. Briefly, different stock solutions of PC (600 mg/mL), Chol (50 mg/mL), TS (50 mg/mL), DDAB (50 mg/mL) and DP (50 mg/mL) were dissolved in an organic phase composed of chloroform and methanol (9:1, *v*/*v*). Three different types of liposomes were prepared: PC, Chol and either TS, DDAB or DP were mixed at a molar ratio of 75:40:5 mM, respectively, in a round-bottomed flask. For BER containing liposomes, 5 mg of BER dissolved in 400 μL of methanol were mixed with the three lipids before the film formation. Organic solvents were evaporated in a rotary evaporator R-300 (Buchi, Flawil, Switzerland) at 40 °C for approximately 20 min, to form a film. The film was finally hydrated with 3 mL of PBS PH 7.4 during 30 min at 40 °C in a rotary evaporator without vacuum. Then, obtained liposomes were sonicated in a MicrosonTM XL 2000 ultrasonic cell disruptor (Misonix, Farmingdale, NY, USA) for 3 min at 8 Watts and stored at 4 °C overnight. To remove the free BER, BER precipitated overnight was discarded and BER liposomes were passed through a PD10 Sephadex^®^ G-25M column (GE Healthcare, Amersham, UK) using PBS as elution buffer. Finally, liposomes, named as PC:Chol:TS, PC:Chol:DDAB or PC:Chol:DP, were lyophilized with trehalose (10% *w*/*v*) and stored at 4 °C until further use.

### 2.6. Characterization of BER Liposomes

BER liposomes were characterized in terms of size, polydispersity and Z potential by photon correlation spectroscopy and electrophoretic laser Doppler anemometry, respectively, using a Zetaplus apparatus (Brookhaven Instrument Corporation, Holtsville, NY, USA). The diameter of the liposomes was determined after their dispersion in distilled water (1:100) and measured at 25 °C with a scattering angle of 90 °C. The zeta potential was measured after dispersion of 2 mg of the formulation in distilled water. Moreover, encapsulated BER was quantified in a UV-visible spectrophotometer at 349 nm after disruption of liposomes with an aqueous solution of 1% (*v*/*v*) Triton X-100. The amount of drug was calculated from a standard curve of BER (0.25 to 15 µg/mL) made in methanol. Lipid content was ascertained using the phosphate assay method, as previously described [28].

### 2.7. In Vitro Cytotoxicity of BER and BER Liposomes

In vitro cytotoxicity assays were performed in BMDM, using MTT. Briefly, 5 × 10^4^ BMDM, were seeded in 96-well plates and incubated at 37 °C for 24 h in supplemented DMEM medium. Then, different concentrations of BER, BER liposomes and AmB were added to each well and plates were again incubated for 48 h. Cells without drug treatment were used as control. After incubation, 20 µL of MTT (5 mg/mL) were added to each well and plates were incubated for 4 h. Then, 100 µL of DMSO were added after removing the medium from each well, and plates were gently shaken for 30 min. Cell viability was determined using a microplate reader (iEMS Reader MS, Labsystems, Bradenton, FL, USA) at 570 nm and the 50% cytotoxic concentration (CC_50_) for each formulation was calculated. Results are expressed as mean ± SD for at least three independent experiments in triplicate wells.

As PC:Chol:DP and PC:Chol:DDAB liposomes were discarded, from now on, PC:Chol:TS liposomes will be named as LP-BER.

### 2.8. Nitric Oxide Production

Nitrites (NO_2_^−^) accumulation in cell culture supernatants of BMDM was used as an indicator of nitric oxide (NO) production and it was determined by the Griess reaction. Briefly, 5 × 10^4^ BMDM were seeded in 96-well plates and incubated at 37 °C for 24 h in supplemented DMEM medium. Then, different concentrations of BER, LP-BER or blank LP (LP, in a concentration equivalent to 25-µM BER in LP-BER), alone or plus 0.1 μg/mL of *Escherichia coli* lipopolysaccharide (*E. coli* LPS, Sigma-Aldrich, St Louis, MO, USA) were added to each well and plates were again incubated for 48 h. Cells without drug treatment were used as control. After incubation, 50 μL of culture supernatants from macrophage culture were dispensed in 96-well plates. Then, 50 μL of a mixture 1:1 of Griess–Ilosvay’s A and B reagents (Panreac, Barcelona, Spain) were added to each well and plates were incubated at room temperature for 30 min. The optical density of the colored product formed was measured on a microplate reader (iEMS Reader MS, LabSystems, Bradenton, FL, USA) at 540 nm. The amount of NO formed in each sample was calculated by comparing them with a standard sodium nitrite (NaNO_2_) concentration curve. Cells incubated with 0.1-μg/mL LPS were use as positive control. Results are expressed as mean ± SD for at least three independent normalized experiments.

### 2.9. In Vitro Antileishmanial Activity of BER and LP-BER Against L. infantum Promastigotes

The leishmanicidal effect of BER and LP-BER was determined in *L. infantum* promastigotes grown in supplemented M199 1X medium by the MTT assay. Briefly, 3 × 10^5^ parasites in exponential phase per well were seeded in 96-well plates with different concentrations of BER. Plates were then incubated at 26 °C for 48 h. Parasites without drug treatment were used as control. AmB was used as positive control. After incubation time, 20 µL of MTT solution (5-mg/mL in PBS) were added to each well and plates were incubated at 26 °C for 4 h. Then, 80 µL of DMSO were added and plates were gently shaken for 30 min. Parasite viability was determined using a microplate reader (iEMS Reader MS, LabSystems, Bradenton, FL, USA) at 570 nm. The concentrations inhibiting the parasite growth by 50 and 90% (IC_50_ and IC_90_, respectively) were calculated with GraphPad Prism7 version (GraphPad Software, Inc., San Diego, CA, USA). Results are expressed as mean ± SD for at least three independent experiments in triplicate.

### 2.10. In Vitro Antileishmanial Activity of BER and LP-BER Against L. infantum Amastigotes: Back Transformation Assay (BTA)

The activity of BER and LP-BER against *L. infantum* amastigotes was also evaluated by the BTA at 48 h, as previously reported [29]. Briefly, 5 × 10^4^ BMDM were seeded in 96-well plates and incubated at 37 °C for 24 h in DMEM medium. Then, cells were infected with stationary promastigotes of *L. infantum* in a proportion 20:1 (parasite:macrophage) and incubated with parasites overnight at 37 °C and 5% CO_2_. After this, cells were washed at least three times with complete DMEM and treated with different concentrations of BER and LP-BER. AmB was used as a positive control. After incubation times, medium was removed, and plates were incubated at 26 °C with complete Schneider’s modified medium to favor the release of remaining viable amastigotes. After 7 days, wells containing medium with remaining parasites were transferred to new 96-well plates and the MTT assay was performed as mentioned above. The selectivity index (SI) of each compound was calculated as the ratio between cytotoxicity (CC_50_) in BMDM and activity (IC_50_) against *L. infantum* amastigotes. Data presented as mean ± SD for at least three independent experiments in triplicate wells.

### 2.11. In Vitro BER Release Study in Serum

The release of BER from LP-BER was evaluated at 25 °C in PBS with a 10% (*v*/*v*) of methanol, to ensure sink conditions. Briefly, 20 mg of LP-BER were dispersed in 1 mL of FBS. The suspension was allowed to dialyze (100 KDa molecular weight cut-off (MWCO)) against 20 mL of PBS 10% methanol in a continuously stirred beaker. At predetermined intervals of time, 1 mL of the buffer was taken, and the same volume of fresh buffer was added to the dialysis. Samples were stored at −20 °C prior to analysis. Total amount of BER released from LP-BER was then quantified measuring the absorbance at 349 nm, as mentioned above. Free BER, dissolved in 10% of methanol, was also dialyzed for comparison. Results are expressed as mean ± SD (n = 3).

### 2.12. Pharmacokinetic Studies of BER and LP-BER

BALB/c mice (n = 9) were treated intravenously (i.v.) or i.p. with free BER or LP-BER at a dose of 7.5 mg/kg (i.v.) or 15 mg/kg (i.p.) of BER. Previously, BER was dissolved in type I water with 5% glucose (*w*/*v*) at 1 mg/mL. Blood was collected from the submandibular plexus of mice at determined time-points: 0.08, 0.5, 1, 2, 4, 8 and 24 h. Plasma samples were obtained by centrifugation of blood at 6000 *g* for 10 min and were kept at −80 °C until analysis. For quantifying, samples were allowed to thaw at room temperature for 10 min. After vortexing, 50 µL of plasma were treated with 400 µL of acetonitrile containing 1% (*v*/*v*) formic acid. Then, samples were vortexed and centrifuged at 6000 *g* for 10 min. Then, 350 µL of each plasma sample containing BER was collected, put into vials and assayed using high performance liquid chromatography with ultraviolet (HPLC/UV). Verapamil hydrochloride was used as the internal standard. Analysis was carried out in a 1200 series LC (Agilent Technologies, Waldbronn, Germany) with a diode-array detector set at 349 nm. The chromatographic system was equipped with a C18 column (75 mm × 3 mm, 2.7-µm particle size; Waters, Milford, MA, USA) operating at 40 °C and sample injection volume was 10 µL. The mobile phase—pumped at 0.7 mL/min—was a mixture of 0.1% (*v*/*v*) formic acid in acetonitrile and 0.1% (*v*/*v*) formic acid in type I water (30:70) using isocratic conditions. For quantification, calibration curves were made in the range of 2.5–500 ng/mL and were prepared using blank plasma. All curves met the previously established performance criteria (R^2^ > 0.999, slopes significantly different from 0 and a relative error (in%) for calibrators <15%). Finally, pharmacokinetic parameters were calculated using Excel Solver program [30]. The following parameters were estimated: half-life (t_1/2_), time of maximum concentration (t_max_), maximum observed plasma concentration (C_max_), rate of clearance (Cl), area under the concentration–time curve (AUC_0–t_) and mean residence time (MRT_0–t_). Furthermore, the relative bioavailability (Fr) of BER or LP-BER after their i.p. administration was estimated by the following Equation (1):Fr (%) = (AUC_i.p._ × Dose_i.v._/AUC_i.v._ × Dose_i.p._) × 100(1)
where AUC_i.v._ and AUC_i.p._ are the areas under the curve for the i.v. or i.p. administrations, respectively. To calculate LP-BER Fr, AUC of LP-BER after their i.v. administration was used. Cl for i.p. route was calculated as (Equation (2)):Cl = Dose × Fr/AUC_i.p._(2)

### 2.13. Tissue Distribution of BER and LP-BER

The amount of BER after i.p. injection of 15 mg/kg of free BER or LP-BER resuspended in PBS was evaluated in spleens and livers of BALB/c mice (n = 3, per time). At 8, 24 and 48 h after drug administration, organs were dissected, weighted and stored at −80 °C before HPLC analysis. BER was extracted from liver and spleen as previously described [31]. Briefly, 20 μL of a solution 100 ng/mL of Verapamil hydrochloride were added (internal standard) and vortexed. A solution containing 100 μL methanol and 300 μL acetonitrile was then added for protein precipitation under vigorous vortex for 5 min and samples were then centrifuged at 18,000 *g* for 15 min. The supernatant was evaporated under vacuum and then reconstituted with 100 μL of the mobile phase. After centrifugation at 18,000 *g*, the supernatant was transferred to a new vial for HPLC/UV analysis, as previously described. The percentages of recovery from liver and spleen samples were 94.6 ± 15.3 and 97.6 ± 16.9%, largely within the 90–110 range considered to be acceptable.

### 2.14. In Vivo Efficacy Studies of BER and LP-BER

Four weeks after i.v. infection of mice with 10^8^ stationary *L. infantum* parasites, i.p. treatments were initiated. Different groups were evaluated (n = 6 per group): untreated control (group 1 for 5 days and group 2 for 10 days of treatment), 5 mg/kg of free BER during 5 consecutive days (group 3), 5 mg/kg of LP-BER during 5 consecutive days (group 4), 10 mg/kg of free BER during 5 consecutive days (group 5), 10 mg/kg of LP-BER during 5 consecutive days (group 6), 15 mg/kg of free BER during 10 consecutive days (group 7), 15 mg/kg of LP-BER during 10 consecutive days (group 8) and FGZ at 1 mg/kg i.v., during 5 and 10 consecutive days (groups 9 and 10, respectively, as positive control). Once treatments were finished, spleen, liver and BM were aseptically removed, and the parasite load was quantified by qRT-PCR.

### 2.15. DNA Extraction and Parasite Quantification

A Macherey–Nagel NucleoSpin^®^ Tissue kit was used as per the manufacturer’s instructions to isolate DNA from the liver, spleen and bone marrow of infected mice. The parasite burden was measured by quantitative real time-polymerase chain reaction (qRT-PCR) of 10 ng of total DNA (quantified by NanoDrop (NanoDrop Technologies ND-1000 UV-vis spectrophotometer)), using the iQ™ SYBR^®^ Green Supermix (Bio-Rad) and specific primers for minicircle kinetoplastic DNA (kDNA) of *Leishmania* (Leish kDNA, Table 1) by the CFX96 real time PCR detection system (Bio-Rad, Hercules, CA, USA). The number of kDNA copies was determined by extrapolation from the cycle threshold of each sample on a standard curve of known concentrations. The standard was generated by insertion of *Leishmania* amplicon in a pCR2.1-TOPO vector (TOPO TA cloning kit; Invitrogen, Carlsbad, CA, USA). Results are expressed as number of copies of the plasmid/10 ng of total DNA.

### 2.16. Biochemical Analysis after BER and LP-BER Administration: In Vivo Toxicity Studies

Blood samples taken at the end of the in vivo efficacy study, after 10 days i.p. treatment with free BER and LP-BER, kept at room temperature for 30 min and then centrifuged at 3500× *g* for 10 min. In order to evaluate the renal and liver toxicity, serum was harvested from each blood sample and alanine aminotransferase (ALT), aspartate aminotransferase (AST), alkaline phosphatase (ALP), total cholesterol (CHO), HDL cholesterol (HDL), LDL cholesterol (LDL), lactate dehydrogenase (LDH), urea (BUN) and triglycerides concentrations (TRIG) were measured in a Cobas^®^ biochemistry analyzer (Roche, Basel, Switzerland). Serum levels of treated mice were compared to untreated ones. Data expressed as mean ± SD (n = 6/group).

### 2.17. Statistical Analyses

Statistical analyses between three groups were performed by using Kruskal–Wallis (nonparametric) followed by Dunn’s multiple comparisons tests. Differences between two groups were analyzed by an unpaired t-test. GraphPad Prism7 version (GraphPad Software, Inc., San Diego, CA, USA) was used to perform the analyses. Significance was established for a *p* value < 0.05.

## 3. Results

### 3.1. Characterization of BER Liposomes

Three different liposomal formulations were prepared by the thin-film hydration method for the encapsulation of BER. The physicochemical characteristics of the different liposomes are presented in Table 2. Liposomes presented sizes that ranged from 120 to 250 nm. PC:Chol:TS liposomes had a mean size of 133.8 nm, and this size did not increase with the encapsulation of BER (121.6 nm). They showed a negative Zeta potential, decreasing with BER encapsulation (−22 mV vs −38 mV) and the obtained loading was 6.6-nmol BER/μmol lipid. On the contrary, liposomes formulated with DDAB (PC:Chol:DDAB) were positively charged (Z-potential around 55 mV) and their mean sizes were 159.6 and 151.4 nm, for blank and BER-loaded liposomes, respectively. However, low encapsulation efficiency (EE) was reached, only 1.3-nmol BER/μmol lipid were loaded. As shown in Table 2, the use of DP in the film formation increased BER encapsulation efficiency to 28.7% (loading = 17.9-nmol BER/μmol lipid). The loading of BER in PC:Chol:DP liposomes importantly increased the mean size of the resulting nanocarriers (245.2 nm for BER-loaded liposomes vs 195.4 nm for blank liposomes) but has negligible effects on their Z-potential and PDi. Thus, PC:Chol:DP liposomes displayed a negative charge of −47 mV. Moreover, all types of liposomes displayed homogeneous distribution (PDi below 0.3) and high lipid yields, around 80–90%.

### 3.2. In Vitro Cytotoxicity of BER and BER Liposomes and NO Production

Previous to the activity studies, cytotoxicity of BER, the three different liposomal formulations and AmB, as positive control, was assessed in BMDM after 48 h (Table 3). AmB, one of the first line treatments for VL, showed the highest toxicity (CC_50_ = 4.5 μM). The cytotoxic effect of free BER was 125.3 μM. However, after its encapsulation into PC:Chol:TS or PC:Chol:DDAB liposomes, this value increased 10-fold, to a CC_50_ higher than 1000 μM (last concentration tested, without any sign of toxicity), indicating an important decrease in toxicity. On the other hand, although the incorporation of BER into PC:Chol:DP liposomes was the highest (17.9-nmol BER/μmol lipid, Table 2), the cytotoxic effect of this formulation was much higher than PC:Chol:TS or PC:Chol:DDAB liposomes (CC_50_ of 140.2 vs >1000 μM, respectively) and similar to the CC_50_ obtained for free BER. Regarding the higher toxicity of PC:Chol:DP liposomes and the very low EE of PC:Chol:DDAB formulation, we decided to continue our experiments with PC:Chol:TS liposomal formulation, from now on named as LP-BER. Regarding NO concentration determined in the supernatants of BMDM, an increase in the NO production was observed after 48 h of treatment with BER or BER encapsulated in liposomes plus LPS (Figure 1), with values higher than the positive control. This NO production increased with increasing concentrations of BER (in presence of LPS). Contrary to what was previously reported [9], BER alone had no effect in NO production. LP-BER added with LPS showed comparable NO production to free BER although, at 5 µM, NO production for LP-BER was 1.6-fold higher than BER. Contrary, blank LP did not influence NO production either alone or plus LPS.

### 3.3. In Vitro Drug Release Study in Serum

In order to study the influence of serum proteins on the stability of liposomes, LP-BER were incubated with 100% FBS. Figure 2 represents the in vitro release profile of encapsulated BER (LP-BER) as cumulative percentage of drug released versus time. Free BER was used as a control of drug diffusion across the dialysis membrane. LP-BER showed a slower drug release, compared to the diffusion of free BER over the first 4 h, followed by a sustained release up to 72 h. One hour after incubation in FBS, only a 15.7% of BER was released from liposomes. On the contrary, 86.3% of free BER was detected (6.4-times higher).

### 3.4. In Vitro Antileishmanial Activity of BER and LP-BER

Table 4 shows the antileishmanial activity of BER, LP-BER and AmB (used as positive control). In *L. infantum* parasites, LP-BER activity was similar to free BER in both promastigotes (IC_50_ = 6.8 ± 0.6 and 5.2 ± 0.1 μM, respectively) and amastigotes (IC_50_ = 1.4 ± 0.2 and 1.1 ± 0.1 μM, respectively). Moreover, although AmB activity against *L. infantum* extracellular promastigotes was higher (4-fold) than free or encapsulated BER, their activity against intracellular amastigotes was similar. Importantly, despite the high AmB activity, a low SI was observed. Free BER SI was much higher than AmB (115.0 vs 3.5, respectively) and LP-BER were much more selective in killing intracellular parasites than macrophages, increasing the SI to a value higher than 714.3 (Table 4).

### 3.5. Pharmacokinetic Studies

As seen in Figure 3, free BER is characterized by a quick disappearance from circulation, either i.v. or i.p. administered. Thus, we analyzed whether liposomal encapsulation was able to modulate the pharmacokinetic profile of BER in vivo after i.p. or i.v. administration (Figure 3). The plasma concentration-time curve clearly indicated that, in comparison with the quick clearance of free BER after injection (Figure 3, ○), liposomal BER displayed an increased retention in blood circulation (Figure 3, ■). Four hours after BER i.v. administration at 7.5 mg/kg (Figure 3a), BER plasma concentration was 4-fold higher when encapsulated in liposomes, compared to the free administered drug (65 vs. 17 ng BER/mL, respectively). However, similar BER levels were observed at 24 h post-administration (20 and 16 ng/mL, for LP-BER and free BER, respectively). When LP-BER were i.p. administered at 15 mg/kg (Figure 3b), BER plasma concentration after 8 h was 13-fold compared to free BER i.p. administered at the same dose (277 vs. 22 ng BER/mL, respectively). Moreover, these differences were maintained and even greater at 24 h (19-fold, 214 vs. 11 ng/mL for LP-BER and free BER, respectively) and BER persisted in the bloodstream for 48 h after its administration in liposomes.

Table 5 summarizes the pharmacokinetic parameters estimated for a noncompartmental analysis of the experimental data obtained. It is noteworthy that LP-BER significantly decreased the Cl of the free drug both by i.v. and i.p. route (9399.8 and 9334.5 mL/h·kg, for free BER after i.v. or i.p. administration, respectively, versus 1716.8 and 1978.8 mL/h·kg, for LP-BER i.v. or i.p. administered). The i.p. route provided higher and sustained plasmatic levels of BER than its i.v. administration (AUC values of 806.8 vs. 504.6 ng/mL·h for free BER and 11,756.7 vs 4253.8 ng/mL·h for LP-BER, respectively). Therein, i.p. LP-BER increased 2.6-fold the Fr value obtained for free BER (62%). Moreover, LP-BER C_max_ value was lower than the obtained by i.v. route (461.5 vs 2565.1 ng/mL, for i.p. vs i.v. administration, respectively).

Moreover, BER accumulation in liver and spleen was measured after i.p. administration of 15 mg/kg of free BER or LP-BER at 8, 24 and 48 h (Figure 4). At 8 h post-administration, BER concentrations were similar in livers and spleens (10.7 and 11.1 ng/mg, respectively) after i.p. administration of LP-BER and much higher than free BER in both organs (107 and 27-fold, for liver and spleen, respectively). Higher levels of BER were also detected at 24 h post-administration of LP-BER vs free BER (47 and 27-fold, in liver and spleen, respectively), and higher BER accumulation was detected in livers (3.3 ng/mg) than spleens (1.9 ng/mg) at 24 h. After 48 h, BER was still detected after LP-BER treatment in these organs.

### 3.6. In Vivo Efficacy Studies of BER and LP-BER

To evaluate the in vivo efficacy (Table 6 and Figure 5) of BER and LP-BER, BALB/c mice were infected i.v. with 10^8^
*L. infantum* parasites. After 4 weeks, treatments were started. Parasite burden in liver, spleen and BM of mice treated i.p. for 5 consecutive days, was similar in mice treated with either 5 or 10 mg/kg of free BER and LP-BER than in non-treated mice (Table 6). Although FGZ was used as positive control for parasite elimination, its administration during 5 consecutive days did not show efficacy in removing *L. infantum* parasites. On the contrary, when FGZ was administered for 10 consecutive days (also at 1 mg/kg), a significant decrease of parasite burden in the liver (*p* < 0.001), spleen (*p* < 0.0001) and BM (*p* < 0.05) was observed (Table 6). Thus, free BER and LP-BER were also administered for 10 days at 15 mg/kg. In this case, parasite burden in the BER group slightly decreased compared to non-treated mice (reduction of 1 logarithm in both liver and spleen). When BER was encapsulated into liposomes, a significant reduction in parasite burden of both liver (Figure 5, *p* < 0.05, reduction of 3 logarithms) and spleen (Figure 5, *p* < 0.01, reduction of 1.3 logarithm) was observed. This parasite reduction was not observed after 10 days treatment with either free BER or LP-BER in the BM (Table 6).

### 3.7. In Vivo Toxicity Studies

The results of the evaluation of the toxicity in vivo, after 10 days i.p. treatment at 15 mg/kg BER, are given in Table 7. There were no significant changes (*p* > 0.05) among the samples for the two transaminase levels (ALT and AST) and the LDH and CHO levels (total CHO, HDL and LDL). Free BER significantly decreased (*p* < 0.01) urea levels (BUN), although these levels are within the normal values reported by Charles Rivers for healthy BALB/c mice [7–31 mg/dL] [32]. However, BER and LP-BER caused a decrease (*p* < 0.05 for LP-BER) in triglycerides out of normal range (TRIG, normal values reported by Charles Rivers for healthy BALB/c mice [101–595 mg/dL]) [32].

## 4. Discussion

Although VL affects million of poor people worldwide, current therapies are far from satisfactory. BER has been reported to have a variety of biologic effects against cancer, diabetes, hypertension, Alzheimer’s disease, cardiovascular diseases or infectious diseases, among others [33,34] and it is currently being tested in clinical trials phase IV for hyperglycemia and metabolic syndrome, what would reduce the costs and time for its use in other therapeutic applications (drug repurposing). Moreover, it was effective against different *Leishmania* species, both in vitro and in vivo [11,12]. However, its use as an antileishmanial drug is hampered by a rapid metabolism and inadequate tissue distribution to target *Leishmania* parasites. After oral or i.v. administration of BER to rats, it is predominantly distributed in the liver and its metabolism occurs primarily in this organ. Several metabolites of BER have been identified in the liver of rats and humans after its administration and subsequent demethylation, glucuronidation and reduction processes. These metabolites include berberrubine, thalifendine, demethyleneberberine, jatrorrhizine, palmatine, columbamine, oxyberberine or dihydroxiberberine, among others [35,36]. Many studies have already revealed that BER metabolites exert similar bioactivities [35]. However, studies in vitro and in vivo carried out with palmatine, jatrorrhizine, oxyberberine or a mixture of jatrorrhizine and columbamine against *L. donovani* strain showed reduced antileishmanial activity compared to BER [12,37]. To overcome these limitations, nanotechnology has been considered as a main strategy [20]. In this study, we prepared common liposomes as a vehicle for BER delivery, aiming to target infected organs.

Injecting drugs encapsulated in liposomes have several advantages compared to free drugs. Due to their colloidal nature, liposomes are recognized as foreign particles and can readily be taken up by phagocytic cells such as macrophages. Consequently, they inherently accumulate in the mononuclear phagocyte system (MPS), preferably, in organs with fenestrated endothelium such as liver, spleen and BM, what constitutes an advantage for the treatment of VL, whose amastigotes harbor inside macrophages of these organs [38]. Moreover, the physicochemical properties of liposomes, such as particle size and membrane charge, are critical issues in the ability of liposomes to target desired tissues and influence the pharmacokinetics and antileishmanial efficacy of liposome-encapsulated drugs. Spherical particles smaller than 200 nm are captured by Kupffer cells of the liver and marginal zone splenic macrophages, whereas particles higher than 200 nm are retained in the red pulp of the spleen [39]. Moreover, nanoparticles between 100–200 nm have been shown to accumulate in the BM [40,41]. Thus, liposomes of 100–200 nm in size have been considered as suitable for their distribution in the three infected organs (liver, spleen and BM). Different methods have been previously used for the obtention of common BER liposomes, such as the pH gradient-film method [31] or ethanol injection method [18], with variable encapsulation efficacies (85 and 10.4%). In this study, we prepared liposomes by film-hydration method and homogenization of their size by sonication. Three different types of liposomes that presented the same proportion of PC and Chol and an anionic or cationic lipid in their composition were prepared (Table 2). Generally, charged particles are more likely to be taken up by macrophages than neutral particles. Moreover, Chol and saturated or cationic lipids have been associated with the induction of proinflammatory macrophage polarization and then, suitable for the preparation of liposomes with indirect antileishmanial activities (macrophage-mediated). Positively charged liposomes have shown antiparasitic activity [42,43,44], which may be attributed to charge-based interactions of cationic liposomes with negatively charged membranes of parasites in addition to their proinflammatory activity. Actually, DDAB cationic liposomes (33.6 mV) containing sodium stibogluconate were reported as effective in killing *L. donovani* parasites in vivo [44]. Although positively charged liposomes have often been associated with cytotoxicity [45], PC:Chol:DDAB liposomes containing BER showed no cytotoxicity in vitro in BMDM (CC_50_ > 1000 µM, Table 3). However, the encapsulation efficiency of BER in these liposomes was very low (2%, Table 2), probably due to the electrostatic repulsions between the positively charged BER molecule and the cationic lipid DDAB, which led us to discard this liposomes composition. DP was used for the preparation of anionic liposomes as the previously reported strong affinity of BER by phosphate groups [46] could increase its encapsulation. Although a higher encapsulation efficiency was reached (28.7%, Table 2), PC:Chol:DP-BER showed much higher toxicity than PC:Chol:TS-BER (>7-fold, Table 3) and similar to free BER, as well as higher size compared to PC:Chol:TS-BER (122 vs 245 nm, Table 2). TS have been described to stabilize liposomes structure comparable to Chol and to inhibit leakage from liposomes [47]. Although electrostatic interactions with lipids affected the loading of BER into the different liposome composition, we hypothesized that the drug would be mainly located in the aqueous core of the liposome, based on its logP of −1.5 [48] and its aqueous solubility (around two milligrams per milliliter [49]). Moreover, this location would better explain the observed BER sustained release profile from LP-BER, prolonged for more than 72 h (Figure 2).

Although BER encapsulation in liposomes formulated with TS (named as LP-BER) could be improved, we decided to continue our studies with this formulation because of its high SI. In fact, the in vitro determination of the 50% inhibitory concentration (IC_50_) against *L. infantum*-infected macrophages was similar for BER and liposome-entrapped BER (IC_50_ 1.1 vs 1.4 µM, Table 4). However, the low toxicity of LP-BER lead to a higher SI for liposomes than BER (6.2-fold increase). The increased NO levels found in vitro (Figure 1) could enhance the therapeutic potential of BER once reaching the organs, inducing a Th1 immune response and leading to a reduction in the parasite load. BER have been described to stimulate the MAPK pathway via inducing the production of NO [9]. Moreover, BER encapsulation in liposomes did not impair this NO production (Figure 1).

In vivo, in a mice model of VL, after i.p. daily administration of 15 mg/kg LP-BER for 10 days, LP-BER significantly reduced the parasitemia in liver and spleen by 3 and 1.3 logarithms (99.2 and 93.5%), respectively (Table 6 and Figure 5), whereas the free drug was only able to decrease the parasite burden in the liver by 1-log. This efficacy improvement can be explained by the higher accumulation of the drug in these organs in mice that received the liposomal formulation (Figure 4). As shown in Table 5, LP-BER have a profound effect in the bioavailability and pharmacokinetic profile of the free drug, probably because free and LP-BER are cleared very differently from the peritoneum. LP-BER would be captured by peritoneal macrophages or cleared via the draining lymphatics since they are too large to cross the blood capillary endothelium. Once taken up by the lymphatic capillaries and passed through regional lymph nodes, they reach the general circulation. On the contrary, BER would be absorbed into the mesenteric vessels which drain into portal vein and pass through the liver, thus, undergoing hepatic metabolism before reaching systemic circulation [50,51]. The result is that LP-BER greatly decrease the Cl of the free drug.

Although LP-BER enhanced drug concentration in liver and spleen compared to the free drug (Figure 4) and showed antileishmanial activity (Figure 5), it is noteworthy that the drug concentration in these organs was much lower than the in vitro IC_50_ values (0.41 μg/mL, Table 4). As the antileishmanial activity of drugs is dependent on drug concentrations in organs where the parasites are located [52] (liver, spleen and BM for VL and skin lesions in the case of CL), this observation is indicating that either BER metabolites or BER immunomodulatory activities (macrophage-mediated effect) are also contributing to the efficacy of LP-BER, reinforcing, on one hand, the idea that several BER metabolites exert similar bioactivities [35], and, on the other hand, the immunomodulatory effect of BER via production of NO [9]. The low plasmatic levels obtained after i.p. or i.v. administration of the free drug (Figure 3) are in agreement with a previously published work, where only a 0.14 and 0.7% of the administered BER was detected after i.p. or i.v. administration, respectively [53]. Thus, these low plasmatic levels (0.3% after i.p. and 0.2% after i.v. administration) provided an extremely high Cl value (Table 5). The pharmacokinetic characteristics of BER as well as its organ accumulation have been much more widely investigated in rats [31,54]. According to Cl value in rats, BER would present an intermediate rate of clearance [31], whereas it would be considered as very rapid in mice. Although LP-BER decreased the clearance of the drug by 10-fold, it was still very high. On the other hand, whereas BER accumulation in organs was higher than plasma levels in rats after oral administration [55], we obtained very lower concentration of BER in mice livers and spleens than in plasma. Our data of BER concentration in these organs were much lower than previously reported with another liposomal BER formulation after its i.v. administration at five milligrams per kilogram in BALB/c nude mice [31]. These liposomes had different composition, showed high drug loading and better sustained release properties. More important, the route of administration was different, as we administered LP-BER by the i.p. route. We chose i.p. administration because it provided higher and sustained plasmatic levels of BER than its i.v. injection (Table 5, AUC_0–t_ of 806.8 vs. 504.6 ng/mL·h, for i.p. vs. i.v. administration, respectively) and overall enhanced the bioavailability (Fr = 160, Table 5) compared to i.v. LP-BER. Moreover, the obtained C_max_ was lower after i.p. administration, and acute cardiac toxicity of BER have been also previously reported to be directly correlated with this parameter [53]. Thus, this administration route would offer better therapeutic interval for the administration of higher doses than by the i.v. route. However, the drug accumulation in the organs after i.v. injection should be determined to decide the most appropriate administration route for LP-BER with regard to VL treatment.

Unfortunately, blood samples taken at the end of in vivo efficacy experiment showed that this schedule of BER and specially LP-BER i.p. administrations significantly reduced the plasma triglycerides levels [32] (Table 7). Although this result confirms the therapeutic potential of BER in hyperlipidemia [56], it could be considered as an off-target or side effect in a VL treatment.

Overall, although liposomal BER delivery enhanced the antileishmanial activity of free drug, their clearance remained very high and had lipid-lowering effect. However, we hypothesized that a liposomal formulation with long-circulating properties (i.e., pegylated liposomes) could decrease BER accumulation in the liver slowing down their clearance and its effect in the lipid metabolism. Moreover, we plan to test the efficacy of the improved formulation with an in vivo protocol of infection more suitable to proof the stability of parasite reduction, leaving a time delay between end of treatment and parasite load quantification.

Thus, we consider that liposomal BER merit further investigations as a strategy to combat VL.

## Figures and Tables

**Figure 1 pharmaceutics-12-00858-f001:**
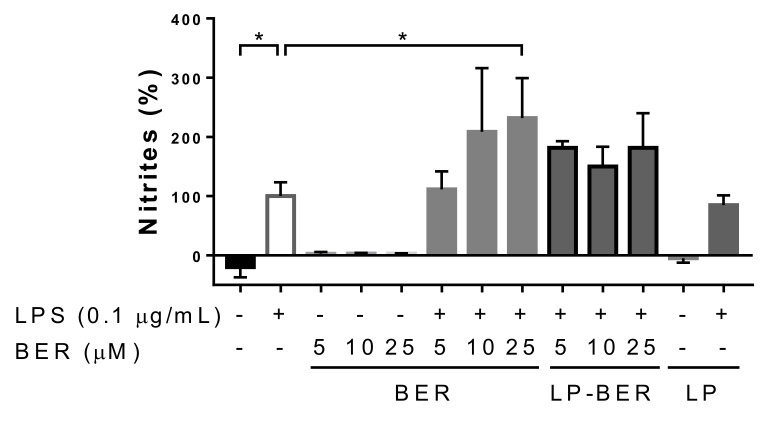
Nitrites production by bone marrow-derived macrophages (BMDM) after 48-h treatment with different concentrations of free berberine (BER), LP-BER (berberine liposomes) or blank LP (LP, equivalent to 25-µM BER in BER LP), alone or plus *Escherichia coli* lipopolysaccharide (LPS) (0.1 μg/mL). Macrophages stimulated with LPS used as positive control. Results expressed as mean ± SD (n = 5). Data analyzed by one-way ANOVA followed by Dunnett’s multiple comparisons test. * *p* < 0.05.

**Figure 2 pharmaceutics-12-00858-f002:**
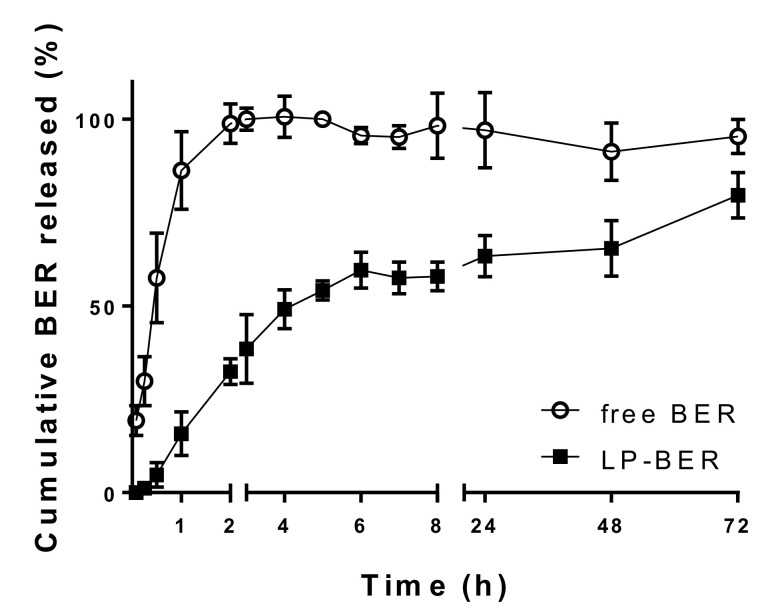
In vitro release of BER from LP-BER using dialysis bag diffusion method in PBS with 10% (*v*/*v*) methanol, at 25 °C. Data presented as mean ± SD (n = 3).

**Figure 3 pharmaceutics-12-00858-f003:**
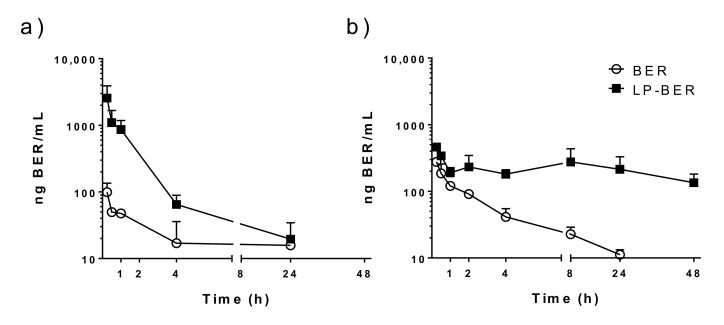
Plasma concentration-time curve of BER in mice after (a) i.v. or (**b**) i.p. administration of free BER and LP-BER at (**a**) 7.5 or (b) 15 mg/kg. Data represents mean ± SD, n = 3 mice per group.

**Figure 4 pharmaceutics-12-00858-f004:**
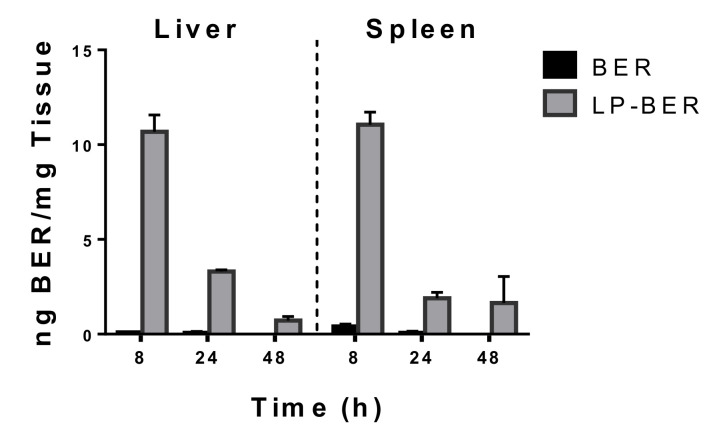
BER accumulation in livers and spleens after 8, 24 and 48 h post i.p. administration of 15 mg/kg of free BER or LP-BER. Data presented as mean ± SD, n = 3 mice per group.

**Figure 5 pharmaceutics-12-00858-f005:**
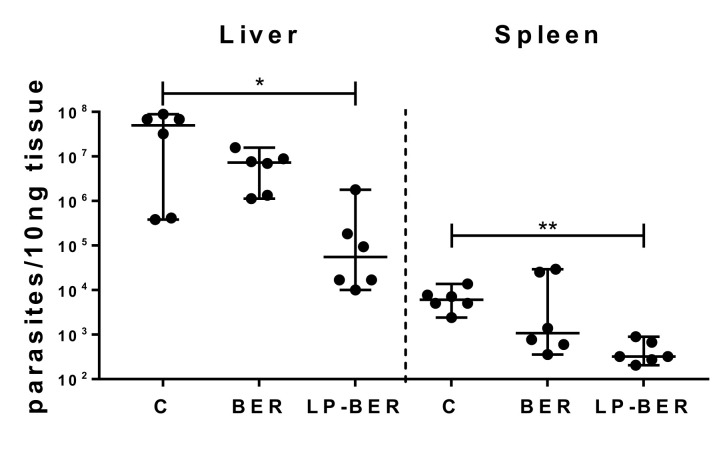
Parasite burden in liver and spleen, measured by qRT-PCR, after 10 consecutive days of i.p. treatment with free BER and LP-BER at 15 mg/kg. Results expressed as median ± 95% IC (n = 6 per group). Data analyzed using a nonparametric Kruskal–Wallis test, followed by Dunn’s multiple comparison. * *p* < 0.05, ** *p* < 0.01.

**Table 1 pharmaceutics-12-00858-t001:** Primers used for parasite burden quantification.

Primer	Sense Primer (5′-3′)	Antisense Primer (5′-3′)
**Leish kDNA**	CCTATTTTACACCAACCCCCAGT	GGGTAGGGGCGTTCTGCGAAA

**Table 2 pharmaceutics-12-00858-t002:** Physicochemical properties of blank and berberine (BER)-containing liposomes.

Liposomes (75:40:5 mM)	Mean Diameter (nm)	PDi	Z-Potential (mV)	BER EE (%)	nmol BER/ μmol lipid
PC:Chol:TS	133.8 ± 1.4	0.19	−22 ± 1	–	–
PC:Chol:TS-BER	121.6 ± 1.3	0.25	−38 ± 1	14.1 ± 2.9	6.6 ± 1.8
PC:Chol:DDAB	159.6 ± 0.9	0.27	52 ± 1	–	–
PC:Chol:DDAB-BER	151.4 ± 0.8	0.27	58 ± 3	2.1	1.3
PC:Chol:DP	195.4 ± 4.8	0.28	−47 ± 1	–	–
PC:Chol:DP-BER	245.2 ± 2.3	0.21	−46 ± 1	28.7 ± 1.4	17.9 ± 3.8

Abbreviations: PDi—polydispersity index; EE—encapsulation efficiency; PC—soybean phosphatidylcholine; Chol—cholesterol; TS—d-α-tocopherol succinate; DP—dicetyl phosphate; DDAB—didodecyldimethylammonium bromide.

**Table 3 pharmaceutics-12-00858-t003:** In vitro cytotoxicity (CC_50_) of free BER, PC:Chol:TS, PC:Chol:DDAB, PC:Chol:DP liposomes or amphotericin B (AmB) in bone marrow-derived macrophages (BMDM) after 48-h treatment.

Drug	BMDM CC_50_ (μM)
BER	125.3 ± 18.4
PC:Chol:TS-BER	>1000
PC:Chol:DDAB-BER	>1000
PC:Chol:DP-BER	140.2 ± 1.6
AmB	4.5 ± 1.2

**Table 4 pharmaceutics-12-00858-t004:** In vitro antileishmanial activity expressed as concentrations inhibiting the parasite growth by 50 and 90% (IC_50_ and IC_90_, respectively) on *L. infantum* promastigotes and amastigotes and selectivity index (SI) in bone marrow-derived macrophages (BMDM) after 48 h of treatment with free BER, LP-BER and amphotericin B (AmB, as positive control).

Drug	Promastigotes (μM)	Amastigotes (μM)	SI (CC_50_/IC_50_)
BER IC_50_	5.2 ± 0.1	1.1 ± 0.1	115.0
BER IC_90_	46.6 ± 0.1	9.8 ± 0.1	–
LP-BER IC_50_	6.8 ± 0.6	1.4 ± 0.2	>714.3
LP-BER IC_90_	60.8 ± 0.6	12.2 ± 0.2	–
AmB IC_50_	1.4 ± 0.3	1.3 ± 0.2	3.5
AmB IC_90_	12.6 ± 0.3	11.4 ± 0.2	–

**Table 5 pharmaceutics-12-00858-t005:** Pharmacokinetic parameters of free BER and LP-BER in BALB/c mice after i.v. or i.p. administration at 7.5 or 15 mg/kg, respectively. Data expressed as mean ± SD (n = 3).

		i.v. 7.5 mg/kg		i.p. 15 mg/kg	
Parameter	Unit	BER	LP-BER	BER	LP-BER
t_1/2_	h	12.9 ± 3.5	4.1 ± 1.4 *	11.7 ± 4.9	16.2 ± 8.2
t_max_	h	0.25	0.25	0.25	0.25
C_max_	ng/mL	99.2 ± 36.0	2565.1 ± 1349.6 *	276.8 ± 29.7	461.5 ± 47.7 **
Cl ^†^	mL/h·kg	9399.8 ± 3148.9	1716.8 ± 882.4 *	9334.5 ± 250.4	1978.8 ± 265.3 ***
AUC_0–t_	ng/mL·h	504.6 ± 213.8	4253.8 ± 1246.1 *	806.8 ± 83.2	11,756.7 ± 976.4 **
MRT_0–t_	h	9.2 ± 1.5	2.2 ± 0.6 **	6.1 ± 0.5	25.3 ± 4.1 **
Fr	%	–	–	62.0	160.0

Abbreviations: t_1/2_—half-life; t_max_—time of maximum concentration observed; C_max_—maximum observed plasma concentration; Cl—rate of clearance; AUC—area under the concentration-time curve; MRT—mean residence time; Fr—relative bioavailability. Results analyzed by unpaired t-test. * *p* < 0.05, ** *p* < 0.01, *** *p* < 0.001 ^†^ Cl for i.p. administration was calculated as dose × Fr/AUC.

**Table 6 pharmaceutics-12-00858-t006:** In vivo efficacy studies in a Leishmania infantum visceral murine model. Parasite burden in liver, spleen and BM was measured by qRT-PCR after 5 or 10 consecutive days of i.p. treatment with free BER and LP-BER at 5, 10 and 15 mg/kg. Fungizone^®^ (FGZ) i.v. administered at 1 mg/kg was used as positive control. Results expressed as median ± 95% IC (n = 6 per group). Data analyzed using a nonparametric Kruskal–Wallis test, followed by Dunn’s multiple comparison. * *p* < 0.05, ** *p* < 0.01, *** *p* < 0.001 and **** *p* < 0.0001.

			Liver		Spleen		BM (Bone Marrow)	
	Dose (mg/kg)		Median	95% IC	Median	95% IC	Median	95% IC
5 Days		Control	3.4 × 10^7^	4 × 10^6^–8 × 10^7^	2.8 × 10^3^	1 × 10^3^–5 × 10^3^	5.6 × 10^5^	3 × 10^5^–1 × 10^6^
	5	BER	4.4 × 10^6^	9 × 10^5^–1 × 10^7^	2.5 × 10^3^	6 × 10^2^–2 × 10^4^	8.7 × 10^5^	2 × 10^5^–3 × 10^6^
		LP-BER	3.5 × 10^7^	1 × 10^7^–1 × 10^8^	2.9 × 10^3^	2 × 10^3^–1 × 10^4^	1.1 × 10^6^	2 × 10^5^–4 × 10^6^
	10	BER	4.9 × 10^7^	2 × 10^7^–1 × 10^8^	2.3 × 10^3^	7 × 10^2^–9 × 10^3^	2.0 × 10^6^	8 × 10^5^–3 × 10^6^
		LP-BER	1.7 × 10^7^	7 × 10^6^–5 × 10^7^	2.5 × 10^3^	2 × 10^3^–6 × 10^3^	6.6 × 10^5^	1 × 10^5^–1 × 10^6^
	1	FGZ	8.0 × 10^5^	3 × 10^5^–1 × 10^7^	6.0 × 10^2^	2 × 10^2^–2 × 10^3^	2.3 × 10^5^	2 × 10^4^–5 × 10^5^
10 Days		Control	5.0 × 10^7^	4 × 10^5^–9 × 10^7^	6.1 × 10^3^	2 × 10^3^–1 × 10^4^	5.1 × 10^5^	8 × 10^4^–2 × 10^6^
	15	BER	7.3 × 10^6^	1 × 10^6^–2 × 10^7^	1.1 × 10^3^	4 × 10^2^–3 × 10^4^	7.9 × 10^5^	1 × 10^5^–3 × 10^6^
		LP-BER	5.6 × 10^4^ *	1 × 10^4^–2 × 10^6^	3.2 × 10^2^ **	2 × 10^2^–9 × 10^2^	1.1 × 10^6^	8 × 10^4^–5 × 10^6^
	1	FGZ	1.8 × 10^2^ ***	1–7 × 10^2^	5.2 ****	1–1 × 10^1^	1.3 × 10^1^ *	6–2 × 10^1^

**Table 7 pharmaceutics-12-00858-t007:** Biochemical analysis after BER and LP-BER administration. In vivo toxicity of BER and LP-BER compared to untreated control mice after 10-days i.p. administration.

Parameter	Control	BER	LP-BER
ALT (U/L)	94.7 ± 23.7	61.0 ± 12.2	60.8 ± 42.9
AST (U/L)	120.6 ± 16.4	105.2 ± 14.1	88.3 ± 24.2
BUN (mg/dL)	41.8 ± 7.3	26.0 ± 4.0 **	29.8 ± 3.0
LDH (U/L)	366.8 ± 55.4	372.5 ± 108.6	457.8 ± 245.3
TRIG (mg/dL)	109.5 ± 15.4	89.1 ± 25.1	70.3 ± 21.6 *
CHO (mg/dL)	114.3 ± 8.5	111.4 ± 4.8	116.8 ± 16.3
HDL (mmol/L)	2.7 ± 0.1	2.6 ± 0.1	2.2 ± 0.3
LDL (mmol/L)	0.4 ± 0.1	0.5 ± 0.1	1.3 ± 0.6

Abbreviations: ALT—alanine aminotransferase; AST—aspartate aminotransferase; BUN—urea; LDH—lactate dehydrogenase; TRIG—triglycerides; CHO—cholesterol; HDL—high-density lipoprotein; LDL—low-density lipoprotein. Results expressed as mean ± SD (n = 6 per group). Data analyzed by Kruskal–Wallis test followed by Dunn’s multiple comparisons test. * *p* < 0.05, ** *p* < 0.01.

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
