# Peer review of "Berberine-Loaded Liposomes for the Treatment of Leishmania infantum-Infected BALB/c Mice"

_pharmaceutics, 2020, doi:10.3390/pharmaceutics12090858_

Round 1
Reviewer 1 Report
Very interesting study. The authors evaluated the potential use of BER-containing liposomes for the treatment of VL, determining their antileishmanial activity in a L. infantum-infected mice model.However, the text lacks the motivation for using the experimental mouse model: the experimental model mimics many of the features of canine and human infections.
Syrian hamsters also exhibit severe clinical signs and symptoms that are similar to those observed in naturally infected dogs and humans.
Reviewer 2 Report
The manuscript of Calvo et al proposes that Berberine chloride (BER - 1,8,13α-tetra-hydro-9,10-demethoxy-2,3-(methyl-ene-dioxy)-berberium chloride) loaded in liposomes (BER-LP) to prevent its rapid liver metabolism and improve this drug selective delivery to the infected organs in visceral leishmaniasis, in a murine model. The manuscript describes the developing and characterization of BER-LP and evaluates its potential use for the treatment of visceral leishmaniasis. The major conclusion of this study was to show the BER immunomodulatory activity or drug metabolites are also contributing to the efficacy, showing a decreased 10-fold rapid clearance of the free drug in mice and reduced plasma triglycerides levels. However, some aspects need the author attentions. My main concerns are the background section and some presented data. The authors are encouraged to improve some aspects of the manuscript prior to publication. Some points that should be addressed prior to publication:
- The introduction is incomplete and sometimes falls outside the focus of the proposed problem. The authors must write about the biological evidence on effects of BER directly on the parasites, amastigotes and promastigotes, and indirectly by immune system stimulation.
- The authors are e encouraged an introducing a drug control for better interpretation of in vivo It is difficult to accept the advantages of BER-LP against L. infantum infection in the absence of in vivo antileishmanial activity control.
- The authors need to review throughout the manuscript the statement concerning the effect of BER on Nitric Oxide production. Data presented in Figure 1 indicated the LPS used is the major inductor of Nitric Oxide production. Perhaps, the way as these results are presented is leading to this interpretation.
- The authors are e encouraged to reviewer statement on the efficacy of BER-LP on the treatment of L. infantum-infected mice model. The results presented correspond to observations at the end of the treatment. With these results, it is only possible to state that the drug's action occurs during treatment. There was no evidence of elimination of the parasite from the tissues, as well, there is no evidence of the prolonged action of BER-LP with a tendency to decrease the parasitic load.
Reviewer 3 Report
The manuscript by Calvo et al. report preparation of Berberine (BER)-loaded liposomes (LPs) and their application for treating leishmaniasis. Different formulation of liposomes were used to prepare BBR-LPs and the size, BBR release profile and biological activities in macrophage of these formulation was tested. Furthermore, the authors also investigated the pharmacokinetics and parasite burden in liver and spleen in Leishmania infantum-infected BALB/c mice and showed that BBR-LP improve drug accumulation and therapeutic effects in vivo. However, the following points should be addressed before considered for publication in Pharmaceutics.
Comments for this manuscript:
- Is BER trapped within the lipid bilayer or the interior aqueous phase? Since the solubility of Isoquinoline alkaloids is generally low, it is most likely that BER is incorporated within the lipid bilayer, but the author should mention this in the manuscript. Furthermore, if BER is in the lipid bilayer, how does this affect stability of liposomes and BER release profiles?
- In Table 2, why did the Z potential of TS LPs decreases after BER encapsulation? Since BER is positively charged, the value after BER encapsulation is expected to be more positive (higher Z potential).
- In Table 2, why did DP LPs show higher drug loading than TS LPs? Both DP (containing a phosphate group) and TS (containing a carboxyl group) should be negatively charged at physiological pH, but TS contains an aromatic ring which would further facilitate the interaction between LPs and BER.
- In Table 3, toxicity of liposomes alone (without BER) should be tested. Why was DP-BER so toxic? Is it due to fast BER release or toxicity of DP liposome?
- In Figure 1, the effect of LP alone should be tested. In addition, the effect of each formulations in cells without LPS treatment should also be tested to show that the observed effect was not induced by the toxicity or contamination of the formulations.
- If BER has anti-inflammatory function as mentioned in Introduction, it should decrease NO production by inhibiting inflammatory responses. Why does it increase NO production in LPS-treated macrophages?
- In Figure 2, the max release for free BER is at around 60%. It should be 100%. In addition, diffusion of free drug through the dialysis membrane seems to be too slow, and the membrane used in this study is not suited for showing the sustained drug release from liposomes. The author should use dialysis membranes with higher MWCO than 1.5 kDa.
- In Figure 3, why is the circulation time after i.p injection of BER-LP so different from that after i.v. injection?
Round 2
Reviewer 2 Report
Two issues need to be better addressed in the manuscript:
(i) One more time, I suggest that the authors need to reviewer statement on the efficacy of BER-LP on the treatment of L. infantum-infected mice model. No substantial evidence of the BER-LP treatment efficacy stability is presented.
(ii) The treatment with drugs (BER and BER-LP) was via "i.p" while the control drug (Fungizone®) was via "i.v". For the correct comparing, both drugs must be administered similarly since the distribution of drugs follows different kinetics.
Round 3
Reviewer 2 Report
In the new version of the manuscript, the authors accepted some of my comments.